# Alteration of Metabolic Pathways in Osteoarthritis

**DOI:** 10.3390/metabo9010011

**Published:** 2019-01-09

**Authors:** Guangju Zhai

**Affiliations:** Discipline of Genetics, Faculty of Medicine, Memorial University of Newfoundland, St. John’s, NL A1B 3V6, Canada; guangju.zhai@med.mun.ca

**Keywords:** osteoarthritis, metabolism, biomarkers, metabolomics

## Abstract

Sir Archibald Edward Garrod, who pioneered the field of inborn errors of metabolism and first elucidated the biochemical basis of alkaptonuria over 100 years ago, suggested that inborn errors of metabolism were “merely extreme examples of variations of chemical behavior which are probably everywhere present in minor degrees, just as no two individuals of a species are absolutely identical in bodily structure neither are their chemical processes carried out on exactly the same lines”, and that this “chemical individuality [confers] predisposition to and immunities from various mishaps which are spoken of as diseases”. Indeed, with advances in analytical biochemistry, especially the development of metabolomics in the post-genomic era, emerging data have been demonstrating that the levels of many metabolites do show substantial interindividual variation, and some of which are likely to be associated with common diseases, such as osteoarthritis (OA). Much work has been reported in the literature on the metabolomics of OA in recent years. In this narrative review, we provided an overview of the identified alteration of metabolic pathways in OA and discussed the role of those identified metabolites and related pathways in OA diagnosis, prognosis, and treatment.

## 1. Introduction

Osteoarthritis (OA) is the most common and disabling rheumatic disease [1], affecting millions of people worldwide [2]. It is the major cause of joint pain, results in substantial morbidity and disability [3], and imposes a great economic burden on society [4]. Despite the high prevalence and societal burden of OA, there is no cure for it yet. While total joint replacement therapy (TJR) is considered by far the most effective treatment for end-stage OA patients [5] and the majority of patients achieve symptomatic improvement following TJR, a meta-analysis including 14 published studies showed that up to 23% of total hip replacement (THR) patients and 34% of total knee replacement (TKR) patients reported an unfavorable long-term post-operative pain outcome [6].

While the loss of articular cartilage is a prominent hallmark of OA, it is now commonly accepted that all the joint structures are affected including subchondral bone, synovium, meniscus in the knee, periarticular ligaments, and adipose tissue, such as the infrapatellar fat pad in the knee [7]. Structural changes of these joint tissues affect the biomechanics and homeostasis of the synovial joint, leading to the impairment of its functional integrity [7,8,9]. As the Osteoarthritis Research Society International (OARSI) defined recently, OA manifests first as an abnormal joint tissue metabolism followed by anatomic and physiologic changes of the joint tissues that can culminate in illness. Thus, the understanding of the biochemical changes and metabolic alteration in OA is of great importance and will facilitate the development of novel targeted interventions to halt or slow down the disease progression [7]. This review describes the metabolic changes that have been reported in OA and discusses the role of those identified metabolites and related pathways in OA diagnosis, prognosis, and treatment.

## 2. Energy Metabolic Pathways

Adenosine triphosphate (ATP) is the primary source of energy in humans and used in innumerable vital metabolic reaction and physiological functions. ATP can be synthesized during the breakdown of molecular fuels including carbohydrates, fats, and proteins by phosphorylation of adenosine diphosphate (ADP) by two types of process: substrate-level phosphorylation that does not need oxygen, and oxidative phosphorylation that requires oxygen. ATP is involved in purinergic signaling and plays a vital role in regulating chondrocyte function and cartilage maintenance and repair [10,11].

Articular cartilage is 2 to 4 mm thick hyaline cartilage. Unlike most tissue, it does not have blood vessels, nerves, or lymphatics. Nutrition of the articular cartilage is provided by diffusion from the synovial fluid. Because of the limited availability of oxygen, chondrocytes mainly depend on anaerobic metabolism to develop, maintain, and repair the extracellular matrix (ECM) of the cartilage [12], and glucose is an important molecular fuel for that.

### 2.1. Glucose Metabolism

The glycolytic pathway converts glucose into lactate when limited amounts of oxygen are available, or pyruvate which then enters the Krebs cycle. The metabolic conversion process involves two key reactions, namely the phosphoglycerate kinase and pyruvate kinase reactions, and produces two ATP molecules. Anaerobic glycolysis occurs at an increased rate in OA-affected chondrocytes [13] so that sufficient ATP can be generated for the repairing process because ATP acts on P2 receptors to promote ECM synthesis [10,14]. The pyruvate kinase M2 (PKM2) gene, which codes for the production of pyruvate kinase, was reported to be upregulated in OA-affected chondrocytes compared to healthy control chondrocytes [15], supporting this. Thus, any alteration of glucose concentration and its distribution between ECM and inside chondrocytes could have an influence on articular cartilage maintenance and repair. Plasma glucose concentration was found to be significantly higher in symptomatic OA patients than in normal controls evaluated in 1026 patients [16]. In another study of 6197 participants, an increase in fasting glucose concentrations by 0.85 mmol/L was associated with an 18% higher risk of hand OA in men [17]. Increasing evidence showed that hyperglycemia-related disorders, such as diabetes, have been associated with OA. A population-based longitudinal study with an age- and sex-matched random sample of 927 men and women aged 40 to 80 years with over 20 years follow-up found that type 2 diabetes was associated with 2.1 times increased risk for TJR after adjustment for age, body mass index (BMI), and other risk factors for OA, and the risk of TJR increased with disease duration and was similar in men and women [18]. More recently, a study with a huge sample size involving 37,353 type 1 diabetes (T1DM) and 1,218,254 type 2 diabetes (T2DM) patients found that T1DM was associated with a 1.4 times increased risk for knee OA and T2DM with 2.75 times increased risk for knee OA. Interestingly, these associations were more pronounced in non-obese than obese persons [19]. In such a disorder, due to either lack of insulin or insulin resistance, glucose cannot be taken up by cells. As a result, glucose concentration inside certain cells decreases whereas the extracellular glucose concentration increases. Therefore, it could be conceivable that decreased glucose levels inside chondrocytes would reduce chondrocyte’s ability to maintain and repair the damaged ECM. However, available data showed that insulin-dependent glucose transporter, GLUT4, was not found to be expressed in chondrocytes [20], suggesting while decreased glucose concentration is seen inside muscle and fat cells, it might not be the case for chondrocytes. In fact, Rosa et al. [21] found that normal human chondrocytes are able to adjust to variations in the extracellular glucose concentration by modulating GLIT-1 synthesis and degradation which involves the lysosome pathway, consistent with an adaptive response to changes in the in vivo environment [22]. However, they found that OA-affected chondrocytes exposed to high glucose were unable to downregulate GLUT-1, as a result, accumulating more glucose inside the chondrocytes and producing more reactive oxygen species (ROS) which is detrimental to ECM [23].

Further, an increased extracellular glucose concentration in the ECM would lead to increased products of advanced glycation end-products (AGEs), which result in pathologic stiffening of cartilage and ECM [24]. A study in diabetic mice knee joints revealed that the levels of the carboxymethyl lysine (CML), one of the AGEs, and matrix metalloproteinase-1 were increased, and the expressions of the cartilage-specific proteins including type II collagen (COL2), transcription factor SOX9, and aggrecan (AGN) were reduced [19]. Our population-based study revealed that AGEs such as CML, methylglyoxal (MG), free methylglyoxal-derived hydroimidazolone (MG-H1), and carboxyethyl lysine (CEL) in blood were not increased in diabetic OA patients compared to non-diabetic OA patients, but the levels of MG and MG-H1 in synovial fluid were significantly higher in OA patients with diabetes than in those without [25]. In addition, we found that the increased MG and MG-H1 in synovial fluid was positively correlated with phosphatidylcholine metabolism [25], which has been found to be associated with OA [26].

Thus, the published data suggest that during OA development the glycolytic metabolic pathway is upregulated to produce more ATP for the cartilage repair process. When there is an elevated glucose concentration, such as in diabetes, chondrocytes cannot adjust but take in more glucose which in return produce more ROS and lead to damage of cartilage. An increased extracellular glucose level also results in an increased production of AGEs which leads to more cartilage damage. 

### 2.2. Tricarboxylic Acid Cycle (TCA)

It is a series of chemical reactions to release stored energy through the oxidation of acetyl-CoA derived from sugar, fat, and proteins into ATP. Global metabolic profiling of human osteoarthritic synovium found that the levels of several tricarboxylic acid cycles (TCA) intermediates were elevated including citrate, *cis*-aconitate, malate, suggesting upregulated TCA pathway in OA [27]. Further, TCA intermediate succinate, as well as glutamine that contributes to the TCA intermediate alpha-ketoglutarate, was also significantly elevated in late disease culture media compared to early disease cultures. The elevated glutamine could also point to a decrease in glycosaminoglycan (GAG) production as glutamine is required for the production of GAG [27]. A study on synovial fluid showed that when compared with the metabolites of the synovial fluid in rheumatoid arthritis (RA) patients, OA patients had higher levels of metabolites involved in both TCA and glycolysis [28]. When examining the metabolites of synovial fluids between early- and late-stage OA, malate levels were significantly higher in the late-stage OA group than those in the early-stage OA group. Other TCA intermediates metabolites including citrate, succinate, and fumarate also showed the same trends [29], suggesting that the upregulation of the TCA pathway is greater in the late-stage of OA, but might also be that TCA intermediates are produced in excess to what can be processed by the OA-affected tissue and thus are exported into the synovial fluid.

In addition, several vitamins provide cofactors for the enzymes involved in these metabolic pathways including cofactors derived from niacin, thiamine, riboflavin, lipoic acid and pantothenic acid. A Japanese study [30] showed that vitamins E, K, B1 (thiamine), B2 (riboflavin), niacin, and B6 were associated with osteophytes after adjustment for age, BMI, and total energy intake, particularly in women. It also showed that vitamins K, B1, B2, B6, and C were associated with minimum joint space width in women. The study suggests that vitamins K, B, and C may have a protective role against knee OA and might lead to disease-modifying treatments, in part via the Krebs cycle. A clinical trial showed that vitamin B complex including thiamine, niacin, and cobalamin significantly reduced knee pain in knee OA patients and the decrease in knee pain was significantly higher in the patients taking vitamin B complex than those in the vitamin E and diclofenac (COX-2 inhibitor) groups [31]. While B vitamins, such as thiamine, riboflavin, and niacin, are important for the Krebs cycle for energy production, their pain reduction effect observed in this clinical trial might be also via other pathways, such as anti-inflammatory pathways or an enhanced effect and/or availability of norepinephrine, as proposed by the authors [31]. In a rat OA model induced by monosodium iodoacetate (MIA), 200 mg/kg of alpha-lipoic acid (ALA) treatment significantly ameliorated cartilage degeneration. ALA could effectively increase the levels of the collagen type II and aggrecan gene expressions and inhibit apoptosis-related proteins expression [32]. Dietary ALA supplementation has also found to prevent synovial inflammation and bone destruction in collagen-induced arthritis mice [33]. 

These data suggest that the TCA pathway is upregulated in OA, most likely to increase the production of ATP needed to repair damaged cartilage. This seems more pronounced in late-stage OA, and vitamins involved in TCA could also play a role, and supplementation of these vitamins might be beneficial to OA patients.

### 2.3. ß-Oxidation Pathway

The ß-Oxidation pathway is the catabolic process by which fatty acid molecules are broken down to provide energy. It is regulated by controlling mitochondrial uptake of fatty acids by the carnitine shuttle. Tootsi et al. [34] studied 70 end-stage OA patients before joint replacement and 82 age-matched controls and found that serum levels of medium- and long-chain acylcarnitines were significantly decreased in OA patients and were associated with OA radiographic severity. Further, this study showed that the acylcarnitines were associated arterial stiffness, suggesting acylcarnitines might play an important role in the link between OA and cardiovascular diseases. We studied metabolic profiles of the synovial fluid samples of 80 end-stage OA patients who underwent total joint replacement and found that there were two distinct groups with one group having significantly lower concentrations of all the acylcarnitines measured in the study, and the group also tended to have a high prevalence of metabolic-related and cardiovascular diseases [35]. While further studies are needed to confirm these findings, these data suggest a significant role of acylcarnitines in OA, at least, in a subgroup of OA patients, which might explain the observed epidemiological association between OA and metabolic-related or cardiovascular diseases. 

An in vitro study of human primary chondrocytes showed that l-carnitine was very effective to stimulate cell proliferation and to induce ATP synthesis. It also showed that l-carnitine enhanced cartilage matrix GAG component production [36]. A rat OA model treated with acetyl-l-carnitine showed a significantly lower of the total pathological score than those without the treatment. Further, the study found that the treatment of acetyl-l-carnitine decreased MMP13 expression in OA chondrocytes but increased COL2 expression [37]. A clinical trial found that l-carnitine supplementation for eight weeks reduced serum inflammatory mediators, such as IL-1ß and MMP-1, and improved pain in knee OA patients [38]. These data suggest that oral l-carnitine supplementation might be a novel nutraceutical for OA, particularly in a subgroup OA patient with low acetylcarnitine levels.

## 3. Lipid Metabolism

Obesity has been a strong risk factor for both the incidence and progression of OA. This is most likely due to excessive joint loading as loss of, at least, 10% of body weight, coupled with exercise, is recognized as a cornerstone in the management of obese patients with OA [39]. However, it is also recognized that the association between obesity and OA could be due to dysregulation of lipid homeostasis [40]. Emerging evidence found that OA was strongly associated with not only obesity but also other metabolic related diseases, such as diabetes, hypertension, and dyslipidemia, lending credence to recognizing metabolic OA as a subtype [41]. A three-year follow-up study of 1,384 Japanese people [42] found that one metabolic syndrome (MS) component (e.g., overweight, hypertension, dyslipidemia, or impaired glucose tolerance) was associated with 2.3 times increased the risk for knee OA incidence. The risk increased to 2.8 times for those with two components and 9.8 for those with three or more components. Similarly, the risk of knee OA progression was also increased with the accumulation of MS components (1.38 to 2.8 times increased the risk for one component to three or more components). A similar association with prevalence of knee OA was also observed in a large Chinese sample with 5764 participants [43]. The Chingford study found that serum high-density lipoprotein (HDL) cholesterol levels were inversely associated with the incidence of the hand radiographic OA in 11-years follow-up [44]. 

By using a metabolomics approach, Williamson et al. [45] studied metabolic profiles of synovial fluid samples obtained from a small group of patients comprising of 10 with OA, 18 with RA, and 11 with traumatic effusions by ^1^H NMR method. They found that OA patients had low levels of saturated triglycerides, and fatty acid chain lengths of the triglycerides were shorter than that for the other groups. Using a targeted metabolomics approach by the LC-MS method, we studied metabolic profiles of both synovial and plasma samples obtained from a total of 197 subjects consisted of knee OA patients, knee OA patients with diabetes, and healthy controls [46]. We found that two specific phosphatidylcholines (PCs), belonging to an ether lipid subgroup of phospholipids that gives rise to inflammatory mediators, such as platelet activating factor (PAF) and signaling molecules, such as eicosanoids, to be associated with knee OA, namely, phosphatidylcholine acyl-alkyl C34:3 (PC ae C34:3) and phosphatidylcholine acyl-alkyl C36:3 (PC ae C36:3). OA patients had significantly lower levels of the two PCs than controls. We also found that diabetic patients had reduced concentrations of these two PCs than controls, and it appeared that the concentration reduction of these two PCs was additive, e.g., knee OA patients with diabetes had lowest concentrations of the two PCs than knee OA patients alone as well as diabetes patients alone. These data suggest that phosphatidylcholine metabolism is altered in OA and the alteration of this metabolic pathway is shared with metabolic related diseases, such as diabetes. Indeed, we found a significant increase of AGEs in synovial fluid of the diabetic patients which was associated with the levels of these two PCs [25]. 

### Phospholipid Metabolism

Kosinska et al. [47] studied the lipid profiles of the synovial fluid samples obtained from 17 patients with early OA, 13 patients with late OA, 18 patients with RA, and 9 controls obtained postmortem with no history of joint disease by a lipidomics approach. They found that the synovial fluid contained several phospholipid classes including phosphatidylcholine (PC), lysophosphatidylcholine (lysoPC), phosphatidylethanolamine (PE), phosphatidylethanolamine-based plasmalogens, phosphatidylglycerol, phosphatidylserine, sphingomyelin, and ceramide; and PC is the predominant phospholipid class, accounting for ~67% of all phospholipids in the synovial fluid. Compared to the median PC concentration in control synovial fluid, the median PC concentration was increased 2.7-fold in early OA synovial fluid, 5.4-fold in late OA synovial fluid, and 3.9-fold in RA synovial fluid. 

We used metabolite ratios as proxies for enzymatic reaction and found that plasma lysoPCs to PCs ratio was significantly increased in knee OA patients compared to controls [26]. When using the optimal cut-off value of 0.09 to classify subjects into two groups in a longitudinal cohort collected in Australia [26], we found that the subjects with the lysoPCs to PCs ratio of equal or greater than 0.09 had 2.3 times increased the risk for TKR due to OA in a 10-year follow-up. Consistently, Castro-Petez et al. [48] also found that the lysoPCs to PCs ratio, particularly lysoPCs with 16 and 18 carbons in their chains to PCs with four double bonds and 36 or 38 carbons, was increased in early OA patients collected in the Netherlands. More recently, we found that the ratio could also predict cartilage volume loss over 2-years follow-up in symptomatic knee OA patients [49]. 

Increased lysoPCs to PCs in OA patients suggest that the conversion of PCs to lysoPCs metabolic pathway catalyzed by phospholipase A_2_ (PLA_2_) is highly active. Pruzanski et al. [50] found that articular cartilage had a higher concentration of PLA_2_ than synovium and OA patients had a higher PLA_2_ concentration than RA patients, suggesting cartilage is the main source of PLA_2_. The drawbacks of the study were that they did not have normal controls and made no distinction among different forms of PLA_2,_ which includes several unrelated families with common enzymatic activity, such as secreted, cytosolic, and lipoprotein-associated PLA_2_ with each having several isoforms. Recently, we investigated the gene expression levels of multiple forms of PLA_2_ in different joint tissues and found that a specific PLA_2_, PLA_2_G5, was significantly increased in human OA cartilage and synovial membrane compared to controls. The increased expression was more pronounced in cartilage than the synovial membrane, consistent with what Pruzanski et al. [50] found. And the overexpression of the PLA_2_G5 was highly correlated with IL-6 which has been associated with knee OA progression [51]. Thus, we speculate that inflammatory process either as a consequence of OA development or initial factor leads to increased activation of the PC to lysoPC metabolic pathway in OA patients. The overactivation of PC to lysoPC conversion pathway results in the release of long-chain polyunsaturated fatty acids, such as arachidonic acid, which leads to the downstream formation of eicosanoids leading to OA joint symptoms, as certain eicosanoids mediate pain. 

## 4. Eicosanoid Pathway

Eicosanoids consisting of the prostaglandins (PG), thromboxanes (TX), leukotrienes (LT), and lipoxins (LX) are signaling molecules synthesized through three major pathways, namely the cyclooxygenase (COX), the lipoxygenase (LOX), and the cytochrome P450 monooxygenase pathways from the precursor of arachidonic acid [52]. Attur et al. [53] examined plasma lipids prostaglandins E2 (PGE2) and 15-hydroxyeicosatetraenoic acid (15-HETE) and found that both of them were significantly elevated in symptomatic knee OA patients compared to non-OA controls. And plasma PGE2 had an Area Under Curve (AUC) of 0.89 from the ROC curve analysis to distinguish symptomatic knee OA patients from non-OA controls. More recently, Valdes et al. [54] examined arachidonic acid, linoleic acid and 20 oxylipins in synovial fluid from 58 knee OA patients and 44 controls and found that levels of three LXs (prostaglandin D2, 11,12-dihydroxyeicosatrienoic acid (DHET), and 14,15-DHET in synovial fluid were associated with knee OA. Of these, the levels of the 11,12-DHET and 14,15-DHET were significantly higher in affected than unaffected knees of people with unilateral disease. Levels of these and 8,9-DHET were also associated with knee OA radiographic progression in the over 3.3 years follow-up of 87 individuals. COX and LOX inhibitors have been used in OA patients for relieving pain, but with variable effects and significant adverse effects. These data suggest the involvement of the eicosanoid pathway in OA, but further studies with systematic approach examining all the metabolites involved in the pathways are needed to identify the most specific metabolites for OA which would help develop more targeted drugs for OA management. 

## 5. Amino Acid Metabolism

### 5.1. Branched Chain Amino Acid (BCAA)

Branched chain amino acids (BCAAs) including valine, leucine, and isoleucine are essential amino acids, meaning that they cannot be produced within the body but have to be taken by diet. They account for approximately one-third of skeletal muscle protein and are important fuels for energy generation. In a study of 199 knee OA patients and 399 controls collected from the UK with a targeted metabolomics approach, we first found that serum BCAA to histidine ratio was significantly associated with knee OA in women [55]. Subsequently, we confirmed the finding in an independent cohort collected from Newfoundland and Labrador, Canada, and extended to the male population [26]. Using ROC analysis, we demonstrated that the ratio had an AUC of 0.76 in discriminating knee OA patients from controls. However, we found that the ratio could not predict the risk for TKR in a longitudinal cohort with 10 years follow-up [26]. This may suggest the BCAA to histidine ratio reflects metabolic processes occurring with OA and could be considered as a diagnostic marker. While the potential mechanism for the observed association remains elusive, an animal OA model of sheep showed that altered BCAA metabolism was observed in the anterior cruciate ligament transaction (ACLT) induced OA model but not others [56], suggesting that the ratio may be associated with anterior cruciate ligament (ACL) injury related knee OA. Evidence suggests that ACL injury is associated with a substantially increased risk for development of future knee OA in the patellofemoral and tibiofemoral joints [57]. A recent clinical trial showed that exercise therapy combined with BCAA supplements would lead to a significant improvement in hip abductor muscle strength of the contralateral side in women with hip OA [58]. Thus, while these observations require further validation, further exploration into the role of BCAA to histidine ratio as a potential tool for monitoring ACL injury patients for OA risk is also warranted.

### 5.2. Arginine

In a study of 138 end-stage knee OA patients and 121 OA-free controls with using a targeted metabolomics approach, we found that the plasma concentration of arginine was reduced by 31% in knee OA patients compared to the controls [59]. The reference interval for plasma arginine concentration in healthy adults is approximately 80 to 120 μM [60]. Similar results were also reported in an Italian study [61] in which the plasma arginine concentration was reduced by 24% in OA patients compared to the age- and sex-matched controls. In addition, the Italian study found that OA patients had a significantly lower arginine to asymmetric dimethylarginine ratio than the controls, suggesting a poor availability of nitric oxide (NO) in the synovial fluid of the OA patients, which may contribute to the progression of OA [61]. NO is a vasodilator produced by conversion of arginine to citrulline by NO synthase [62]. Increased flux through this pathway would result in increased NO formation. However, the role of NO in the development of OA is still inconclusive with some studies suggesting a protective role whereas others showing a destructive role by mediating the inflammatory response and apoptosis and inhibiting the synthesis of collagen and proteoglycan [63]. Further, in a study of an ACLT OA model of rabbit, the concentration of arginine was decreased after ACLT and the post-ACLT arginine concentration was negatively associated with the disease severity [64], suggesting that the decreased arginine concentration is likely due to OA process and progression, and supplementation of arginine might be beneficial to OA patients. 

## 6. Other Metabolic Factors

Loeser et al. [65], studied urinary metabolic markers and progression of knee OA in overweight and obese adults. They found that glycolate, hippurate, and trigonelline in urine were the important metabolites for distinguishing knee OA progressors from non-progressors assessed by radiography. Hippurate and trigonelline are potentially gut flora-derived metabolites, suggesting that differences in the gut microbiome could be contributing to metabolic differences associated with knee OA progression. 

## 7. Summary

During the initiation and development of OA, energy metabolic pathways including glycolytic and TCA pathways are upregulated to produce ATP needed for cartilage repair. These upregulated pathways might also produce important repair molecules, such as signaling molecules, nucleotides, lipids, and amino acids that need to be studied. Diseases, such as diabetes, affecting these pathways play a significant role in OA, which could be due to increased ROS and AGEs which lead to further damage of articular cartilage (Figure 1). Alteration of the carnitine shuttle process has an impact on beta-oxidation and energy production in OA, at least, in a subgroup of OA patients. This might also be a link shared between OA and metabolic-related and cardiovascular diseases. Phosphatidylcholine metabolism, especially the conversion pathway of phosphatidylcholine to lysophosphatidylcholine catalyzed by the PLA_2_ enzyme, could be a novel target for developing new drugs for OA. Overactivation of this pathway could provide precursors, such as arachidonic acid and upregulate the downstream eicosanoid pathway (Figure 1). Further, several amino acids, particularly BCAA and arginine, as well as vitamins involved in the above-mentioned pathways could provide novel nutraceuticals for OA management. Clinical trials need to be conducted to vet their disease-modifying effects on OA.

In conclusion, the metabolomic studies of OA reported in the literature are promising; these altered metabolic pathways, once confirmed, could provide us new insights into the pathogenesis of OA and help develop new interventions for OA. Given that OA is recognized as a multifactorial and heterogenous disease, metabolomics could help classify OA patients into subtypes [35] so that more targeted interventions could be developed and applied. 

## Figures and Tables

**Figure 1 metabolites-09-00011-f001:**
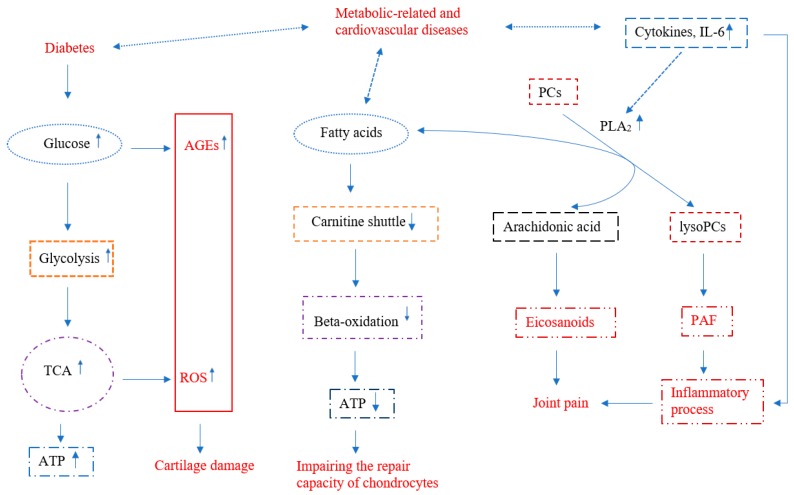
Chart of the metabolic alterations in osteoarthritis (OA) reviewed in this article. Arrows with dash line indicates possible relationships. TCA—tricarboxylic acid cycle; ATP—adenosine triphosphate; PC—phosphatidylcholine; lysoPC—lysophosphatidylcholine; PAF—platelet activating factor; PLA_2_—phospholipase A_2_; IL-6—interleukin 6.

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
