# Peer review of "Alteration of Metabolic Pathways in Osteoarthritis"

_metabolites, 2019, doi:10.3390/metabo9010011_

Round 1

Reviewer 1 Report

Line 27, the clause “Even though,   ” is awkward, consider rewording or just deleting.

Line 47-48, consider using ‘that’ instead of ‘which, and adding a comma.  “ – substrate-level phosphorylation that does not need oxygen, and oxidative phosphorylation that requires oxygen.”

Line 55, hyphen seems out of place.

Line 58, consider rewording.  “… kinase reactions, and produces two ATP molecules.”

Line 62, consider rewording. “…and inside chondrocytes could have an influence on articular cartilage maintenance and repair.”

Line 67, the sentence should start with “A population-based…”

Line 99, consider adding a comma before ‘which’

Line 143, should be “Tootsi et al studied 70 end-stage…”

Line 175, should be “…odds ratio…”

Line 212, should be “… OA patients compared to controls.”

Line 445, update the citation if it is no longer “under review.”

Author Response

I thank the reviewer very much for taking much of his/her time to review the manuscript and provide very constructive suggestions. I have revised the manuscript accordingly and believe that the revised manuscript has been improved greatly. The point by point responses to the comments are as follows:

The manuscript is a well-written review of metabolic pathway alterations that occur in osteoarthritis. The language is often quite informal, for example Lin 100 the paragraph begins with "So, it can be summarized that ...".

GZ: Thanks for pointing out this. We have rephrased the sentence to "Thus, the published data suggest ...". please see page 3, line 103 to 104.

Minor suggestions include:

Line 27, the cause "Even though, " is awkward, consider rewording or just deleting.

GZ: We have revised the first paragraph of the manuscript accordingly. Please see line 26 to 30 on page 1.

Line 47-48, consider using ‘that’ instead of ‘which, and adding a comma.  “ – substrate-level phosphorylation that does not need oxygen, and oxidative phosphorylation that requires oxygen.”

GZ: We have made these changes accordingly. Please see line 48 - 49 on page 2.

Line 55, hyphen seems out of place.

GZ: We have changed it to colon. Please see line 57 on page 2.

Line 58, consider rewording.  “… kinase reactions, and produces two ATP molecules.”

GZ: We have added "and". Please see line 60 on page 2.

Line 62, consider rewording. “…and inside chondrocytes could have an influence on articular cartilage maintenance and repair.”

GZ: We have added "an" before influence and changed it to "repair". Please see line 65-66 on page 2.

Line 67, the sentence should start with “A population-based…”

GZ: We have added "A" before "population-based...". please see line 70 on page 2.

Line 99, consider adding a comma before ‘which’

GZ: We have added a comma before which. Please see line 102 on page 3.

Line 143, should be “Tootsi et al studied 70 end-stage…”

GZ: We have modified accordingly. Please see line 149 on page 4.

Line 175, should be “…odds ratio…”

GZ: According to this reviewer and the editor's comments, we have now rephrased several sentences in the paragraph. Odds ratio has now changed to "times increased risk". Please see line 180 - 183 on page 4.

Line 212, should be “… OA patients compared to controls.”

GZ: We have modified accordingly. Please see line 219 on page 5.

Line 445, update the citation if it is no longer “under review.”

GZ: That paper is still under review and I will inform the journal as soon as it has been accepted and make changes accordingly.

Reviewer 2 Report

This review discusses metabolic changes which occur in OA-afflicted joints vs. health joints - and is structured by pathway: Energy producing pathways (Glycolysis, TCA cycle, fatty acid oxidation), Lipid synthesis pathways (phospholipid and eicosanoid), Amino acid metabolism (branched-chain amino acids and arginine), and finally potential gut-microbiota related metabolism. The author also highlights various co-morbidity effects which have some overlapping relevance to the pathway.

Major comments:

- The overall structure (focusing each section on pathways) was not as effective as it could have been because some sections included information that was tangential to the pathway, even if that tangential information was interesting - examples of this: discussing vitamins in the TCA cycle, diabetes and AGEs in the glycolysis section, or eicosanoids in the phospholipid section. The author is trying to 1) make a case for co-morbidities that might be related to OA (cardiovascular disease, obesity, metabolic syndrome, and diabetes), but also 2) describe OA-associated changes in metabolites within specific metabolic pathways. Because of the 2 objectives, it’s hard to follow the logic, and some of the ideas get mixed between the sections. The microbiota paragraph at the end seems out of place.

- The author takes the stance that increased flow of glucose and increased abundance of TCA cycle intermediates is driven by need of ATP for tissue repair. It could also be that metabolic cycles are needed to produce important repair molecules (nucleotides, lipids, amino acids, signaling molecules), but this isn’t addressed at all by the author.

- Line 34 is almost verbatim from a cited paper (citation #8) “impair[ing] the functional integrity of the synovial joint, adversely affecting its biomechanics and attenuating its [already] limited inherent capacity for repair and regeneration” original paper cited 2 other articles (PMID: 27652499 & PMID: 22265263). In addition the statement in the current manuscript “the disease manifests first as an abnormal joint tissue metabolism, followed by marked alteration in the functional properties…” is a somewhat inaccurate interpretation of the cited manuscript, which says that metabolism is important in OA progression, but might/might not be the first change.

- Line 51 is direct from cited paper (citation #9) “Without a direct supply of nutrients from blood vessels or lymphatics, chondrocytes depend primarily on anaerobic metabolism”

- Several parts of the writing were not accurate.  For example: line 108 - “levels of several TCA intermediates were elevated, including lactate, succinate, and glutamate” - succinate is a TCA intermediate, lactate and glutamate are not TCA intermediates.

Line 45: “ATP can be synthesized from molecular fuels including carbohydrate, fats and proteins…” - Break down of molecular fuels like carbohydrates, fats, and amino acids lead to the production of ATP from ADP through substrate level phosphorylation or oxidative phosphorylation, synthesis of ATP from these fuels is more round-about and goes through purine synthesis pathways.

- Line 58: “Anaerobic glycolysis occurs at an increased rate in OA-affected chondrocytes (10), *so that sufficient ATP can be generated for the repairing process*” - there is no cited evidence that ATP is directly for the repair process, it could be, but the authors in (10) also postulate increased glycolysis could be used to support increased ROS scavenging. The author of this review could be more clear that this is his interpretation of the cited data.

Minor comments:

- typos. Lines 63, 83, 102, 143, 168

- Poor grammar in end of the first paragraph (Lines 26-29), seems like an incomplete sentence.

- Lines 91-93. Idea is not clear, consider rephrasing.

- Line 246. Need to clarify if the AUC is for a predictive model ROC curve.

Author Response

I thank the reviewer very much for taking much of his/her time to review my manuscript and provide very constructive comments/suggestions. I have revised the manuscript accordingly and believe the revised manuscript has been improved greatly. The point by point responses to the reviewer's comments are as follows:

This review discusses metabolic changes which occur in OA-afflicted joints vs. health joints - and is structured by pathway: Energy producing pathways (Glycolysis, TCA cycle, fatty acid oxidation), Lipid synthesis pathways (phospholipid and eicosanoid), Amino acid metabolism (branched-chain amino acids and arginine), and finally potential gut-microbiota related metabolism. The author also highlights various co-morbidity effects which have some overlapping relevance to the pathway. 

Major comments:

- The overall structure (focusing each section on pathways) was not as effective as it could have been because some sections included information that was tangential to the pathway, even if that tangential information was interesting - examples of this: discussing vitamins in the TCA cycle, diabetes and AGEs in the glycolysis section, or eicosanoids in the phospholipid section. The author is trying to 1) make a case for co-morbidities that might be related to OA (cardiovascular disease, obesity, metabolic syndrome, and diabetes), but also 2) describe OA-associated changes in metabolites within specific metabolic pathways. Because of the 2 objectives, it’s hard to follow the logic, and some of the ideas get mixed between the sections. The microbiota paragraph at the end seems out of place.

GZ: Thanks for pointing out this. We feel that metabolic pathway oriented structure for the review would be helpful which would allow us to discuss those epidemiological associations with OA in the context of the relevant metabolic pathways such as diabetes in glycolytic pathways and TCA cycle and obesity and lipid profiles and phospholipid in the lipid metabolic pathways. However, we totally agree with the reviewer that the eicosanoids pathway shouldn't be in the phospholipid section. We have therefore modified accordingly and now the Eicosanoids pathway is an independent section. We also agree with the reviewer that the microbiota paragraph is out of place and we have now deleted that part from the manuscript. Please see changes on page 6 for Eicosanoid pathway and page 7 for section 6.

- The author takes the stance that increased flow of glucose and increased abundance of TCA cycle intermediates is driven by need of ATP for tissue repair. It could also be that metabolic cycles are needed to produce important repair molecules (nucleotides, lipids, amino acids, signaling molecules), but this isn’t addressed at all by the author.

GZ: Thank the reviewer very much for pointing out this which is a very important point and worth to be studied. I have pointed out this in the summary and stated it needs to be studied in the future. Please see line 314 to 316 on page 7.

- Line 34 is almost verbatim from a cited paper (citation #8) “impair[ing] the functional integrity of the synovial joint, adversely affecting its biomechanics and attenuating its [already] limited inherent capacity for repair and regeneration” original paper cited 2 other articles (PMID: 27652499 & PMID: 22265263). In addition the statement in the current manuscript “the disease manifests first as an abnormal joint tissue metabolism, followed by marked alteration in the functional properties…” is a somewhat inaccurate interpretation of the cited manuscript, which says that metabolism is important in OA progression, but might/might not be the first change.

GZ: Thanks for pointing this out. We have now rephrased the statement and added two other references as suggested. please see line 33-35 on page 1 and new references 8 and 9. Further, we have now clarified that the statement of "the disease manifests first as an abnormal joint tissue metabolism...." was the OARSI OA definition. Please see line 35-37 on page 1. 

- Line 51 is direct from cited paper (citation #9) “Without a direct supply of nutrients from blood vessels or lymphatics, chondrocytes depend primarily on anaerobic metabolism”

GZ: We have now rephrased it. Please see line 53 to 54 on page 2.

- Several parts of the writing were not accurate.  For example: line 108 - “levels of several TCA intermediates were elevated, including lactate, succinate, and glutamate” - succinate is a TCA intermediate, lactate and glutamate are not TCA intermediates.

GZ: Thanks for pointing this out. We have now corrected it. Please see line 111 - 114 on page 3.

Line 45: “ATP can be synthesized from molecular fuels including carbohydrate, fats and proteins…” - Break down of molecular fuels like carbohydrates, fats, and amino acids lead to the production of ATP from ADP through substrate level phosphorylation or oxidative phosphorylation, synthesis of ATP from these fuels is more round-about and goes through purine synthesis pathways.

GZ: Thanks. We have now added this on line 49-50 on page 2.

- Line 58: “Anaerobic glycolysis occurs at an increased rate in OA-affected chondrocytes (10), *so that sufficient ATP can be generated for the repairing process*” - there is no cited evidence that ATP is directly for the repair process, it could be, but the authors in (10) also postulate increased glycolysis could be used to support increased ROS scavenging. The author of this review could be more clear that this is his interpretation of the cited data.

GZ: Thanks. We have now made it clearer and added evidence as new references 10 and 14. Please see line 61-62 on page 2.

Minor comments: 

- typos. Lines 63, 83, 102, 143, 168

GZ: All corrected. please see the relevant text in the manuscript. All changes are highlighted.

- Poor grammar in end of the first paragraph (Lines 26-29), seems like an incomplete sentence. 

GZ: We have now modified the first paragraph accordingly. Please see line 26 -30 on the page 1. 

- Lines 91-93. Idea is not clear, consider rephrasing. 

GZ: We have now rephrased it. Please see line 93 to 94 on page2 and line 95 - 96 on page 3.

- Line 246. Need to clarify if the AUC is for a predictive model ROC curve.

GZ: Yes, we have now made it clearer. Please see line 252 on page 6.

Round 2

Reviewer 2 Report

The review has been significantly improved. Still there are a few points that could be addressed.

Line 46: “ATP can be synthesized from molecular fuels” – I still think the word choice ‘synthesized from’ is not the best. Consider – “ATP can be synthesized during the breakdown of molecular fuels…”

Line 53: “In the absence of the direct supply of nutrients from the blood vessels or lymphatics, chondrocytes mainly depend on anaerobic metabolism…” – I know this comes from a cited source, but to me it seems illogical. Aren’t nutrients also needed for anaerobic metabolism? I think the limiting factor is likely oxygen availability more than nutrient availability. The statement is especially illogical when you end it with “glucose is an important molecular fuel for that” (isn’t glucose a nutrient? If nutrients are limited by lack of blood supply, wouldn’t glucose also be limited?... in actuality the synovial fluid is rich in many nutrients but does lack red blood cells).

Line 56: Consider changing title to “glucose metabolism” – this section includes discussion of glucose availability and import, as well as AGEs, which are more generally glucose metabolism vs. glycolytic pathway.

Line 57: Suggest removing “’the end product:”

Line 105: “can’t” to cannot

Section TCA: I think there is some subtlety here that is missing. My interpretation is that TCA in synovial fluid actually suggests that TCA intermediates are produced in excess to what can be processed by the tissue and thus are exported into the synovial fluid. This also suggests increased production of TCA intermediates in tissues, but it doesn’t necessarily mean the system is optimized for greater ATP production and might even suggest that flux of metabolites into TCA cycle is overwhelming the ETC leading to export of the intermediates.

Line 149: Need to specify that the study is measuring serum levels of acyl carnitines.

Line 218: Need to specify that these are ratios in plasma.

Author Response

We once again thank the reviewer very much for taking his/her time to re-review our manuscript and provided further comments. We have considered the comments/suggestions carefully and revised the manuscript accordingly. The point by point responses are as below. We believe that the manuscript have further improved significantly. All the changes are highlighted in the revised manuscript for your convenience.

Line 46: “ATP can be synthesized from molecular fuels” – I still think the word choice ‘synthesized from’ is not the best. Consider – “ATP can be synthesized during the breakdown of molecular fuels…”

GZ: Thanks. We have revised it as suggested. Please see page 2, line 46 -47.

Line 53: “In the absence of the direct supply of nutrients from the blood vessels or lymphatics, chondrocytes mainly depend on anaerobic metabolism…” – I know this comes from a cited source, but to me it seems illogical. Aren’t nutrients also needed for anaerobic metabolism? I think the limiting factor is likely oxygen availability more than nutrient availability. The statement is especially illogical when you end it with “glucose is an important molecular fuel for that” (isn’t glucose a nutrient? If nutrients are limited by lack of blood supply, wouldn’t glucose also be limited?... in actuality the synovial fluid is rich in many nutrients but does lack red blood cells).

GZ: This is a good point, many thanks! we have now modified the sentence and specified it is because of the limited availability of oxygen. Please see page 2, line 54.

Line 56: Consider changing title to “glucose metabolism” – this section includes discussion of glucose availability and import, as well as AGEs, which are more generally glucose metabolism vs. glycolytic pathway.

GZ: We have now changed the subtitle as suggested. please see page 2, line 57.

Line 57: Suggest removing “’the end product:”

GZ: We have now deleted the words as suggested. please see page 2, line 58.

Line 105: “can’t” to cannot

GZ: We have now modified it. please see page 3, line 106.

Section TCA: I think there is some subtlety here that is missing. My interpretation is that TCA in synovial fluid actually suggests that TCA intermediates are produced in excess to what can be processed by the tissue and thus are exported into the synovial fluid. This also suggests increased production of TCA intermediates in tissues, but it doesn’t necessarily mean the system is optimized for greater ATP production and might even suggest that flux of metabolites into TCA cycle is overwhelming the ETC leading to export of the intermediates.

GZ: Thanks, this is an excellent point. We have now added it in the manuscript. Please see page 3, line 123-126.

Line 149: Need to specify that the study is measuring serum levels of acyl carnitines.

GZ: We have now clarified it as serum levels. Please see page 4, line 155.

Line 218: Need to specify that these are ratios in plasma. Please see page, line.

GZ: We have now clarified it as plasma ratios. Please see page 5, line 223.